# *LMM24* Encodes Receptor-Like Cytoplasmic Kinase 109, Which Regulates Cell Death and Defense Responses in Rice

**DOI:** 10.3390/ijms20133243

**Published:** 2019-07-02

**Authors:** Yue Zhang, Qunen Liu, Yingxin Zhang, Yuyu Chen, Ning Yu, Yongrun Cao, Xiaodeng Zhan, Shihua Cheng, Liyong Cao

**Affiliations:** Key Laboratory for Zhejiang Super Rice Research and State Key Laboratory of Rice Biology, China National Rice Research Institute, Hangzhou 310006, China

**Keywords:** lesion mimic mutants (LMMs), programmed cell death (PCD), defense responses, receptor-like cytoplasmic kinases (RLCKs), rice (*Oryza sativa* L.)

## Abstract

Lesion mimic mutants are excellent models for research on molecular mechanisms of cell death and defense responses in rice. We identified a new rice lesion mimic mutant *lmm24* from a mutant pool of *indica* rice cultivar “ZhongHui8015”. The *LMM24* gene was identified by MutMap, and *LMM24* was confirmed as a receptor-like cytoplasmic kinase 109 by amino acid sequence analysis. The *lmm24* mutant displayed dark brown lesions in leaves and growth retardation that were not observed in wild-type ZH8015. The results of histochemical staining and TUNEL assays showed enhanced ROS accumulation and cell death in *lmm24*. Chloroplast degradation was observed in *lmm24* leaves, with decreased expression of photosynthesis-related genes and increased expression of the senescence-induced *STAYGREEN* (*SGR*) gene and other senescence-associated genes. Furthermore, *lmm24* exhibited enhanced resistance to rice blast fungus *Magnaporthe oryzae* (*M. oryzae*) and up-regulation of defense response genes. Our data demonstrate that *LMM24* regulates cell death and defense responses in rice.

## 1. Introduction

Lesion mimic mutants (LMMs) display spontaneous lesions in the absence of pathogen attack, environmental stress, or mechanical damage [1], exhibiting a similar phenotype to the pathogen infection-induced hypersensitive response (HR) mediated through programmed cell death (PCD) [2]. Programmed cell death plays an important role in the innate immunity of plants by initiating the cell death process to defend against pathogen infection [3]. In fact, most LMMs show increased resistance to pathogens: The *spl30* mutant shows enhanced resistance to the bacterial pathogen *Xanthomonas oryzae* pv *oryzae* (*Xoo*); *dj-lm* shows enhanced resistance to rice blast (*M. oryzae*) [4]; and *oscul3a* shows enhanced resistance to both *M. oryzae* and *Xoo* [5,6]. These findings suggest that LMM genes are involved in regulating the plant defense response. Cloning and characterization of LMM genes will likely provide new insights into complex defense pathways in plants.

Lesion mimic mutants have been reported in various plants, such as Arabidopsis (*Arabidopsis thaliana*) [7], maize (*Zea mays* L.) [8], rice (*Oryza sativa* L.) [9], wheat (*Triticum aestivum* L.) [10], and barley (*Hordeum vulgare* L.) [11]. In recent years, more and more *LMM* genes with diverse pathogenesis of lesions have been isolated and characterized in rice. For example, *NLS1* encodes an ancient class of CC-NB-LRR-type resistance proteins, with single amino acid substitutions that lead to constitutive activation of defense responses [12]; the *SPL11* gene encodes a U-Box/ARM protein, and loss of activity of this protein results in out of control PCD [13]; *HPL3* encodes a hydroperoxide lyase, where loss of enzyme function reduces wound-induced green leaf volatile (GLV) emission but increases jasmonic acid (JA) accumulation, producing disease-resembling lesions spread throughout the entirety leaves [14]; and monocot-specific receptor-like kinase SDS2 positively regulates plant cell death and immunity in rice where plants over-expressing *SDS2* show cell death on leaves and accumulate higher levels of ROS [15]. Light also plays an important role in the development of lesion mimics in LMMs: the mutant *NPR1-OX* exhibited a lesion phenotype when grown under low-intensity light conditions (12 h day and night cycle, 28 °C, 90% relative humidity) [16].

Receptor-like cytoplasmic kinases (RLCKs) in plants belong to the super family of receptor-like kinases (RLKs), which lack extracellular ligand-binding domains but show homology to RLKs in kinase domains. Previously, 149 and 379 RLCK encoding genes were identified from Arabidopsis and rice, respectively [17,18]. According to sequence homology, Arabidopsis and rice RLCKs are classified into 17 subgroups and are referred to as RLCK-II and RLCK-IV–RLCK-XIX [19]. Receptor-like cytoplasmic kinases play important roles in diverse developmental processes in plants. By associating with immune receptor kinases (RKs), RLCKs regulate multiple downstream signaling nodes to trigger defense responses against microbial pathogens. OsRLCK176 interacts with chitin elicitor receptor kinase 1 (OsCERK1) and functions downstream of OsCERK1 in the peptidoglycan (PGN) and chitin signaling pathways [20]. Lines expressing RNAi constructs against *OsRLCK57*, *OsRLCK107*, and *OsRLCK118* result in a burst of ROS and induction of pathogenesis-related (PR) genes by chitin and PGN [21]. *Xa21* encodes a receptor-like protein kinase consisting of 1025 amino acids that mediates resistance to *Xoo* by identifying signaling molecules. Silencing of OsRLCK102 affects *Xa21*-mediated resistance to *Xoo*, indicating that OsRLCK102 regulates *Xa21*-mediated immunity [22]. OsRLCK185 transmits OsCERK1-mediated chitin signaling to activate a MAPK cascade of immune responses to regulate rice resistance to *M. oryzae* [23]. Receptor-like cytoplasmic kinases also play a role in sexual reproduction [24], stomatal patterning [25], brassinosteroid signaling [26], plant adaptation to abiotic stresses [27], and other developmental processes [28].

MutMap is a method of rapid gene isolation based on whole-genome sequencing [29]. By resequencing the whole genome of a mutant strain, the researchers can obtain all the sequence differences of the mutant genome from the wild-type genome where the only expected differences are related to the mutant phenotype [30]. With MutMap, researchers use a mutagen to mutagenize a rice cultivar to obtain a mutant parent. This mutant is then crossed with a wild-type (WT) parental line. The F_1_ plants are self-pollinated to generate an F_2_ population; and DNA from multiple recessive mutant F_2_ progeny are used for sequencing analysis. The results of sequencing the progeny pool are compared with that of the wild-type pool, and the different sites are used to identify the gene locus associated with the mutant phenotype [31]. In recent years, this technique has been successfully applied for gene mapping in rice: the rice grain size gene *WTG1* was cloned by MutMap [32]; Wang et al. (2018) identified a regulator of endosperm development in rice by a modified MutMap method [33]; and Fekih et al. (2013) applied MutMap+ for identification of the *Hit9188* mutant with pale-green and dwarf phenotypes [34].

In this study, we identified a lesion mimic mutant, named *lmm24*, from the mutant pool of the cultivar ZhongHui8015. The *lmm24* mutant exhibited spontaneous lesions in leaves. Through physiological and biochemical analyses, we found abnormal ROS accumulation and cell death and early leaf senescence in *lmm24*. Moreover, *lmm24* exhibited enhanced resistance to *M. oryzae.* Finally, we successfully cloned the *LMM24* gene using the MutMap method and found that it encodes the receptor-like cytoplasmic kinase 109.

## 2. Results

### 2.1. Characterization of the lmm24 Mutant

The growth of *lmm24* was significantly weaker than that of ZhongHui8015 (ZH8015) starting at the seedling stage (Figure 1A) and proceeding throughout the duration of growth of plants. Dark brown lesions began to appear in the lower leaves of *lmm24* at the seedling stage (25 days post-sowing (dps)) in the field in summer. Individuals of *lmm24* grown in the greenhouse showed lesions at 20 dps (28 °C, 8 h dark, 16 h light), which then spread from the leaf tip to the entire leaf. However, prior to 25 dps, the leaves of the mutant had no lesions (similar to ZH8015 leaves). Moreover, the newly emerging leaves showed no lesions but gradually developed lesions as the plant developed (Figure 1B,C). There were fewer lesions on the flag leaf at the maturation period, which may be related to the temperature and level of illumination of the environment (Figure 1E).

Numerous studies on LMMs have shown that light is an important factor affecting the occurrence of lesions. In order to determine whether lesion formation in *lmm24* was affected by light, we conducted an antiglare assay on both ZH8015 and *lmm24*. The same size of aluminum foil was used to cover newly emerging leaves of the mutant and ZH8015 to prevent light exposure (Figure 1D). In *lmm24*, areas without exposure to light did not have lesions compared to areas exposed to light. There were no differences between the two parts in ZH8015. Thus, we determined that *lmm24* develops lesions in a light-dependent manner.

In addition, compared with ZH8015, plant height, tiller number, grain size, 1000 grain weight, and seed setting rate of *lmm24* were significantly reduced (Figure 1F,G and Appendix A).

### 2.2. Abnormal ROS Accumulation and Cell Death in lmm24

We speculated that the lesion phenotype of LMMs is the result of ROS accumulation leading to cell death. Thus, we investigated ROS accumulation and cell death in *lmm24* leaves. Evans blue solution can enter cells undergoing death but cannot enter living cells, so Evans blue staining is a histochemical indicator of cell death. The reagent 3,3′-diaminobenzidine (DAB) was used to indicate the accumulation of hydrogen peroxide (H_2_O_2_). Histochemical staining results showed that *lmm24* had many Evans blue spots after staining, whereas ZH8015 did not (Figure 2A), indicating mass cell death in *lmm24*. Comparing DAB staining between the mutant and ZH8015, we found that *lmm24* had much stronger H_2_O_2_ accumulation than ZH8015 (Figure 2B). Terminal deoxynucleotidyl transferase dUTP nick end labelling (TUNEL) assay is another method to detect cell death. When the cell dies, the DNA breaks and the exposed 3’-OH can be added to fluorescein by the TUNEL reagent, which can be detected by fluorescence microscopy. Our TUNEL assay carried out to detect DNA fragmentation in *lmm24* and ZH8015 indicated stronger TUNEL signals were detected in the leaves of *lmm24* (Figure 2C), which also suggests that there was much more cell death in *lmm24*.

### 2.3. Early Leaf Senescence in lmm24

A common feature of most LMMs is that leaf senescence is a consequence of uncontrolled PCD. The degradation of chloroplasts is a critical indicator of leaf senescence, so we compared the ultrastructure of chloroplasts in *lmm24* and ZH8015 using transmission electron microscopy (TEM). Chloroplasts were well-developed with rich lamellae and a small number of osmiophilic bodies in ZH8015 (Figure 3A–C), whereas chloroplast membranes were broken and the thylakoid lamellae had loosely arranged mesophyll cells surrounding the lesion areas in *lmm24* (Figure 3D–F). Expression levels of photosynthesis-related genes (*psaB*, *psbA*, *psbB*, *psbC*, *cab2R*, *rpoA*, *CHLI*, *CHLD*) were significantly down-regulated in *lmm24* mutant leaves (Figure 3G), while expression levels of the senescence-induced *STAYGREEN* (*SGR*) gene and senescence-associated genes (*Osl2*, *Osl30*, *Osl43*, *Osl85*) were significantly up-regulated (Figure 3H). At the same time, *Ubiquitin* was used as a reference gene to analyze the relative expression levels of the above genes, and the results were consistent with the results of *β-actin1* as a reference gene (Appendix A). Overall, these results suggest that early leaf senescence occurs in *lmm24*.

### 2.4. lmm24 Shows Enhanced Resistance to M. oryzae

Previous studies indicated that some LMMs show enhanced resistance to fungal and bacterial pathogens. To determine whether *lmm24* also confers higher resistance to pathogens, we employed the leaf spraying (seedling stage) and punch inoculation (tillering stage) methods to infect ZH8015 and *lmm24* plants with the virulent *M. oryzae* isolate 12-144-1-1 (Figure 4A,B). Seedlings of both the mutant and ZH8015 showed susceptibility, but the morbidity rate of ZH8015 was more serious than that of the mutant. ZH8015 was obviously susceptible at the tillering stage while the mutant was minimally susceptible to disease. The statistical results of lesion area in the two growth stages showed that ZH8015 had significantly higher areas with lesions than the mutant (Figure 4C,D), indicating that *lmm24* enhanced resistance to *M. oryzae*. To determine the mechanism of enhanced resistance of *lmm24*, we investigated the transcription level of PR genes (*PR1a*, *PBZ1*, *PR1b*, *PAL1*, *AOS2*, *WRKY45*) in ZH8015 and *lmm24* (Figure 4E,F); these genes were all significantly upregulated in *lmm24*. The same experiment was performed using *Ubiquitin* as a reference gene. Expression levels of PR genes were also significantly greater in *lmm24* than in ZH8015 (Appendix A). Together, these results suggested that the *lmm24* mutant gene triggered a defense response to *M. oryzae*, which led to enhanced disease resistance associated with lesion formation in *lmm24*.

### 2.5. Genetic Analysis and Use of the MutMap Method to Clone the LMM24 Gene

To isolate the potential causal gene of the observed mutant phenotype, we crossed *lmm24* with ZH8015. F_1_ plants were self-pollinated to obtain F_2_ progeny. All F_1_ plants exhibited a ZH8015 phenotype while the F_2_ progeny exhibited a ZH8015/mutant ratio of 3:1 (Appendix A), suggesting that the phenotype of *lmm24* was controlled by a single recessive nuclear gene.

The MutMap method was used to map the *lmm24* gene. From the F_2_ population, 30 individual plants with mutant phenotypes were selected to extract DNA which were pooled (pool-M) for genome-wide resequencing. We obtained 22,155,026,400 clean base pairs from pool-M and 6,754,683,300 clean base pairs from ZH8015 pool (pool-WT). Compared with the reference genome, the mapping rate of pool-M was 97.78% and the average depth was 50×; the mapping rate of pool-WT was 97.69% and average depth was 16× (Appendix A and Appendix A). These results indicate that the quality and depth of sequencing sufficient for the subsequent SNP analysis.

We used the UnifiedGenotyper module of the GATK3.3 (Broad Institute of MIT and Harvard, Cambridge, MA, USA) software to detect multiple sample SNPs, filtered using VariantFiltration, and annotated SNP results with ANNOVAR (Columbia University, New York, NY, USA). According to the method described by Akira Abe et al. [29], we calculated the SNP index of each SNP and plotted the SNP distributions of 12 chromosomes in rice (Appendix A). Regions with a SNP index of 1 are considered candidate regions for the *lmm24* mutant phenotype. Fortunately, only one region on chromosome 3 was selected as a candidate, corresponding to the rice gene *BGIOSGA012679* (*LOC_Os03g24930*) (Figure 5A). We sequenced the cDNA of *LOC_Os03g24930* and found two nucleotide mutations (C^1009^-to-T, C^1055^-to-T) and a 54 bp insertion; these two loci were in the fourth exon of this gene leading to amino acid changes (Figure 5B). In order to verify the 54 bp insertion, we extracted the DNA of 15 individual plants with mutant phenotypes in the F_2_ group and carried out agarose gel electrophoresis using a molecular marker at the insertion site (4th Exons F/R, Appendix A). The results showed that the 54 bp insertion was present in all 15 individual plants (Figure 5C). Thus, *LOC_Os03g24930* is a candidate gene for *LMM24*.

### 2.6. Functional Complementation with LOC_Os03g24930 in the lmm24

To verify that *LOC_Os03g24930* is the *LMM24* gene, the vector COM-*lmm24* containing a 5192 bp genomic fragment of ZH8015, consisting of the 1842 bp upstream promoter and 1166 bp downstream terminator of *LOC_Os03g24930*, was constructed. This vector was then introduced into *lmm24* by *Agrobacterium tumefaciens*-mediated transformation. Of 32 regenerated T_0_ plants, 27 were positive transformants, and all of them were similar in phenotype to ZH8015 and thus the mutant phenotype of *lmm24* was restored. There were no lesions on the leaves (Figure 6A–C). We performed histochemical staining experiments on the leaves of T_1_ plants. DAB staining results showed that the T_1_ positive transgenic plants, such as ZH8015, had little accumulation of H_2_O_2_ in the leaves. Evans blue staining showed that there was less cell death in the leaves of T1 transgenic plants compared with *lmm24*. Thus, we determined that mutations on the fourth exon of *LOC_Os03g24930* likely caused the mutant phenotype of *lmm24*.

### 2.7. LMM24 Encodes Receptor-Like Cytoplasmic Kinase 109 in Rice

As annotated on RGAP (http://rice.plantbiology.msu.edu), *LOC_Os03g24930* (*LMM24*) is composed of four exons and three introns, a genomic DNA (gDNA) sequence of 2185 bp, a coding sequence of 1344 nucleotides, and encodes a putative 448 amino acid protein with a tyrosine protein kinase domain. Previous research showed that *LMM24* encodes receptor-like cytoplasmic kinase 109 (OsRLCK109) [18]. RLCKs in plants belong to the super family of receptor-like kinases (RLKs), which lack extracellular ligand-binding domains. The predicted protein scheme for OsRLCK109 shows a kinase domain (Figure 7A); 108–393 amino acids encode the kinase domain. Multiple series of alignments revealed that this sequence is highly conserved (Figure 7B), and that the *LMM24* mutation site is in this conserved sequence.

### 2.8. Subcellular Localization of OsRLCK109

To determine the subcellular localization of OsRLCK109, we fused OsRLCK109 with GFP, under the control of the constitutive cauliflower mosaic virus 35S promoter, to obtain the construct *35S::RLCK109-GFP*. Then, we transformed the *35S::RLCK109-GFP* plasmid into rice protoplasts and co-transformed an *35S::Ghd7.1-CFP* plasmid expressed in the nucleus as a marker [35]. We observed the fluorescent signals under a confocal microscope and found green fluorescent signals in both cytoplasm and nucleus where the signal in the nucleus was stronger than in the cytoplasm. This stronger GFP signal was co-localized with the nucleus marker Ghd7.1-CFP (Figure 8). These results suggest that the OsRLCK109 protein was localized mainly in the nucleus but there is also a weaker signal in the cytoplasm.

## 3. Discussion

### 3.1. LMM24 is the OsRLCK109 Gene and Functional Mutation of OsRLCK109 Leads to the Mutant Phenotype

Many LMMs have been identified in rice and have been widely applied to the study of PCD and defense responses. In this study, we isolated an LMM, *lmm24*, from the mutant pool of *indica* rice “ZhongHui8015”. Dark brown lesions spontaneously formed in *lmm24.* Moreover, compared with ZH8015, mutant plants were shorter and had fewer tillers, a later growth period, and lower yield (Figure 1 and Appendix A). Like most LMMs, the lesions mainly appeared on the leaves and the growth of the plant was affected; however, the lesions of *lmm24* differed from those of most LMMs because production of lesions were light-dependent (Figure 1). The *LMM24* gene was identified as *LOC_Os03g24930* by MutMap, which encodes OsRLCK109. Sequencing results showed that the replacement of two single bases and a 54 bp insertion occurred on the fourth exon of *LMM24*, resulting in the change of two amino acids in the kinase domain of OsRLCK109. Mutagenesis by EMS often results in single-base mutations, with fewer examples of large fragment insertions and single-base mutations. The functional complementation of OsRLCK109 restored the mutant to the wild-type phenotype, indicating that the functional mutation of OsRLCK109 is responsible for the mutant phenotypes.

### 3.2. Cell Death and Early Leaf Senescence are Caused by Functional Mutation of OsRLCK109

The *lmm24* mutant displayed spontaneous cell death from the seedling stage to the yellow mature stage. Experimental results show accumulation of H_2_O_2_ in *lmm24*. H_2_O_2_ is a major by-product of beta-oxidation and acts as a signal molecule in the promotion of cell death, thus, excessive accumulation of H_2_O_2_ is the main cause of lesion formation. Like most LMMs, *lmm24* accumulates H_2_O_2_ only on leaves. Unfortunately, more research is needed to find the cause. Early leaf senescence also occurred in *lmm24* and we found chloroplast degradation in lesion areas of *lmm24* by TEM as well as down-regulation of photosynthesis-related genes and up-regulation of the *SGR* gene and senescence-associated genes. In previous studies on early leaf senescence, senescence and lesions were described as two different traits; only a few studies have described early leaf senescence as one of the characteristics of LMMs [6,9]. We believe that lesions are typical of early leaf senescence and that leaf senescence in LMMs usually happens rapidly, providing a unique tool for understanding leaf senescence.

### 3.3. OsRLCK109 Regulates Defense Response in Rice from Disease

The defense response is activated in LMMs to increase broad-spectrum disease resistance to pathogens. We found that *lmm24* improved resistance to rice blast by inoculation experiments and that PR genes were all significantly upregulated. These results suggest that the defense response is activated in *lmm24*, but the molecular mechanism of this resistance is unclear. *LMM24* encodes OsRLCK109 and, reportedly, RLCKs play a key role in the transmission of immune signals, by associating with immune RKs. Pattern recognition receptors (PRRs) located on the plasma membrane are required to perceive the conserved pathogen-associated molecular patterns (PAMPs) during plant innate immunity. Chitin elicitor receptor kinase 1 (OsCERK1) is a type of PRR; the RLCK family members OsRLCK185, OsRLCK176, OsRLCK57, OsRLCK107, and OsRLCK118 were all reported to be located at the plasma membrane because of their interactions with OsCERK1. However, in our study, OsRLCK109 was located in the nucleus and cytoplasm, so we speculated that OsRLCK109 is involved in an immune pathway different from these RLCKs, but further experimental evidence is needed to support our inference.

In conclusion, functional mutation of OsRLCK109 induces spontaneous lesions in leaves and enhances defense responses in *lmm24*. Our results suggest that OsRLCK109 may play an important role in regulating cell death and defense responses in rice. Recent advances suggest a key role of RLCKs in RK-mediated signaling. Correspondingly, most RLCKs are localized to the cell membrane. In this study, RLCK109 was mainly located in the nucleus, which differs from findings of previous studies and may provide new avenues to further investigate the role of RLCKs in the immune signaling pathway.

## 4. Materials and Methods

### 4.1. Plant Material and Growth Conditions

The *lmm24* mutant was isolated from a mutant pool of *indica* rice ZH8015 by ethyl methane sulfonate (EMS) treatment. A cross was made between *lmm24* and ZH8015, the first filial generation (F_1_) was self-pollinated, and the second generation (F_2_) was used for genetic analysis. The *lmm24* and ZH8015 seeds were grown in a plant growth chamber (14 h light/10 h dark, 28/25 °C, 90% relative humidity) for an inoculation experiment with rice blast. All F_2_ and parent populations were grown in an experimental paddy field at the China National Rice Research Institute (Hangzhou, Zhejiang province in China) from May to November, 2017.

### 4.2. Mapping of LMM24

MutMap was performed to map the *LMM24* gene according to a previous study [29]. We generated an F_2_ population from a cross between *lmm24* and ZH8015. DNA from the 30 plants with *lmm24* phenotypes from this F_2_ population was pooled using the same amount of DNA from each plant for whole genome sequencing. DNA purity and integrity were analyzed by agarose gel electrophoresis, and DNA purity was determined by Nanodrop2000 (Thermo Fisher Scientific, Waltham, MA, USA). The qualified DNA samples were then randomly broken into fragments of 350 bp in length by a Covaris crusher. The library was built using the TruSeq Library Construction Kit (Illumina, San Diego, CA, USA), and the constructed library was sequenced by an Illumina HiSeqTM PE150 (Illumina, San Diego, CA, USA). The *indica* 93–11 (http://www.rise.genomics.org.cn) genome sequence was used as a reference genome sequence and the SNP/InDel index was calculated according to a previous report [36]. cDNA of *LOC_Os03g24930* with a two SNP index of 1 was PCR amplified using primers LMM24F/R. The product was subsequently sequenced to confirm the mutation.

### 4.3. Complementation of LMM24

A 5192 bp genomic fragment of ZH8015 was PCR amplified using primer pairs P-LMM24 for complementation of the *lmm24* mutant phenotype. This fragment consisted of an 1842 bp upstream promoter (Appendix A), a 2184 bp gene region, and a 1166 bp downstream terminator. The PCR product was recombined in the binary vector pCAMBIA1300 using an In-Fusion Advantage Cloning kit (Clontech, San Francisco, CA, USA). The resultant expression construct was transformed into *lmm24* by *Agrobacterium tumefaciens*-mediated transformation. The transformation experiment was completed by Wuhan Boyuan Biotechnology Co., Ltd (Wuhan, China).

### 4.4. Histochemical Marker Staining Assay

Leaves (second from the top) of field-grown *lmm24* and ZH8015 at 50 dps were used in this assay. Evans blue staining for dead cells and DAB staining for H_2_O_2_ accumulation were conducted as previously described [37,38].

### 4.5. TEM and TUNEL Assay

Samples of ZH8015 and *lmm24* leaves (second from the top) at 50 dps were prepared for TEM. The leaf segments were fixed in a 2.5% glutaraldehyde solution at 4 °C overnight. The samples were rinsed three times, each time for 15 min, with 0.1 M, phosphate buffer at PH 7.0. The samples were then fixed with 1% osmic acid solution for 1–2 h, dehydrated in a graded ethanol series and embedded in Spurr’s resin. Ultrathin samples were made as previously described [39,40]. In order to identify nuclear DNA fragmentation, which is also a distinct feature of PCD, we used the TUNEL assay; the test method is based on the report of Ku et al. [41].

### 4.6. RNA Isolation and qPCR Analysis

Total RNA was isolated from leaves (second from the top) of ZH8015 and *lmm24* at 50 dps with the TIANGEN RNAprep Pure Plant Kit (TIANGEN, Beijing, China). cDNA was synthesized with a ReverTra Ace kit (TOYOBO, Osaka, Japan) and qPCR was performed on a Roche LightCycler 480 (Roche, Basel, Switzerland) device using a SYBRPremix ExTaq kit (Takara, Dalian, China). The ratio of A260/A280 was 1.8–2.2, and two clear bands of 18s RNA and 28s RNA were distinguishable by agarose gel electrophoresis. Three biological replicates and three technical replicates were performed for each sample. The expression level of the target genes was normalized to that of the *β-actin1* gene and *Ubiquitin* gene (Appendix A). The results of qPCR were statistically analyzed using the 2^–ΔΔCt^ method. In addition, the values of ZH8015 were normalized to clearly show the change of *lmm24* relative to ZH8015.

Plant materials were grown in the experimental field of the China Rice Research Institute. The leaves were immediately immersed in liquid nitrogen and brought back to the laboratory for storage at −80 °C. DNA was removed using DNase for RNA extraction and DNA was removed again using the ReverTra qPCR gDNA Remover kit before reverse transcription. The cDNA was diluted twice.

### 4.7. Multiple Sequence Alignment

Gene information was determined using the Rice Genome Annotation Project database (RGAP, http://rice.plantbiology.msu.edu/). Protein prediction of *lmm24* was conducted using the Simple Modular Architecture Research Tools (SMART) program (http://smart.embl-heidelberg.de/). Homologous sequences of *lmm24* were obtained using the NCBI Blastp search program (http://www.ncbi.nlm.nih.gov/) and multiple sequence alignments were conducted using the software programs MEGA v7 (http://www.megasoftware.net/) and GENEDOC2.7 (https://www.softpedia.com/get/Science-CAD/GeneDoc.shtml).

### 4.8. Subcellular Localization

The full-length CDS of *LMM24* was amplified by GFP-F/R and the PCR product was cloned into the C-terminus of the green fluorescent protein (GFP) coding region in pJIT163-hGFP vectors. Plasmids *35S::RLCK109-GFP* and *35S::GFP* were transformed into rice protoplasts prepared from leaf tissues using the polyethylene glycol (PEG)-mediated transformation method. A *35S::Ghd7.1-CFP* plasmid was co-transformed as a marker. Fluorescence of GFP was detected using a laser confocal scanning microscope ZEISSLSM 700 (ZEISS, Jena, Germany) 48 h after transfection. GFP can be excited by the 488 nm laser line and is optimally detected at 600 nm. ECFP (enhanced cyan fluorescence protein) was excited at 405 nm and was optimally detected at 550 nm. Mercury lamp was used to excite the fluorescent proteins.

### 4.9. Pathogen Infection

For disease resistance evaluation, leaves of ZH8015 and *lmm24* were inoculated with virulent *M. oryzae* pathotypes following a previously described procedure [42].

## Figures and Tables

**Figure 1 ijms-20-03243-f001:**
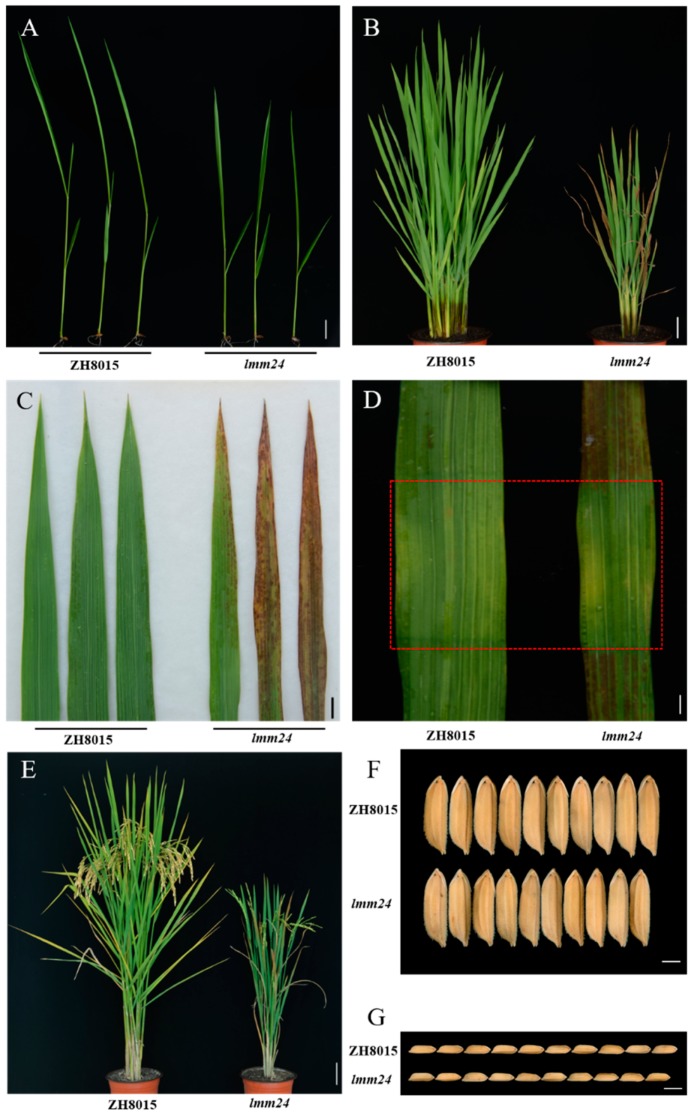
Phenotypic analyses of *lmm24.* (**A**) Comparison of growing situation ZH8015 and *lmm24* at seedling stage (15 days post-sowing (dps)). (**B**) *lmm24* plant at tillering stage (50 dps) display dark brown lesions. (**C**) Lesion mimic phenotype in *lmm24* and clean leaf blade in ZH8015. (**D**) Determination of whether the *lmm24* phenotype was light dependent. The area covered with aluminum was indicated by red rectangle frame. (**E**) ZH8015 and *lmm24* plants at harvest period (**F**) Grain width observation of *lmm24* and ZH8015. (**G**) Grain length observation of *lmm24* and ZH8015. Scale bar: 2 cm in (**A**,**C**), 1 cm in (**D**,**F**,**G**), 10 cm in (**B**,**E**).

**Figure 2 ijms-20-03243-f002:**
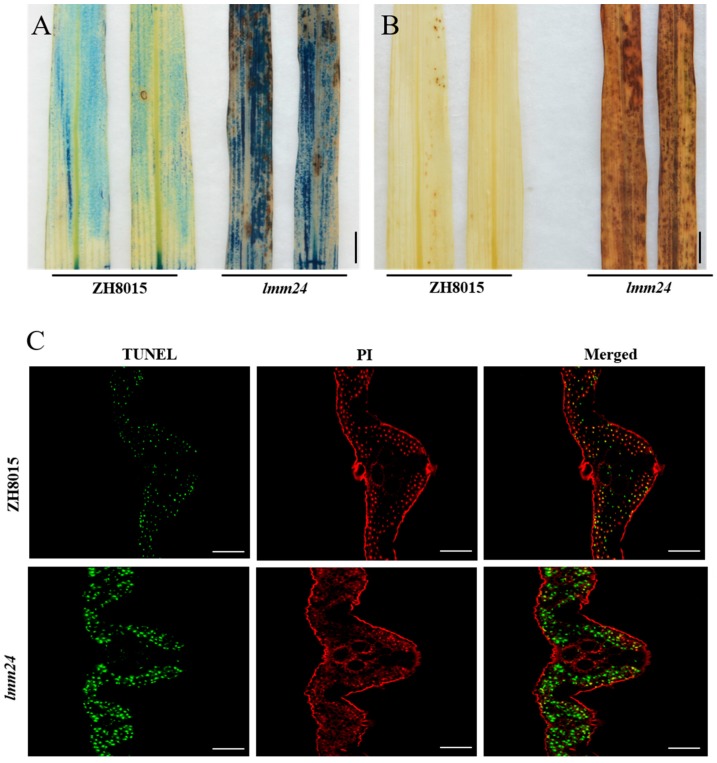
H_2_O_2_ accumulation and cell death detection in ZH8015 and *lmm24*. (**A**) Evans blue staining for cell death. (**B**) Diaminobenzidine (DAB) staining for H_2_O_2_ accumulation. (**C**) DNA fragmentation detection in mesophyll cells by TUNEL assay. Red signal represents staining with propidium iodide, and green signals indicate TUNEL-positive nuclei of dead cells. Scale bar:1 cm in (**A**,**B**), 50 μm in (**C**).

**Figure 3 ijms-20-03243-f003:**
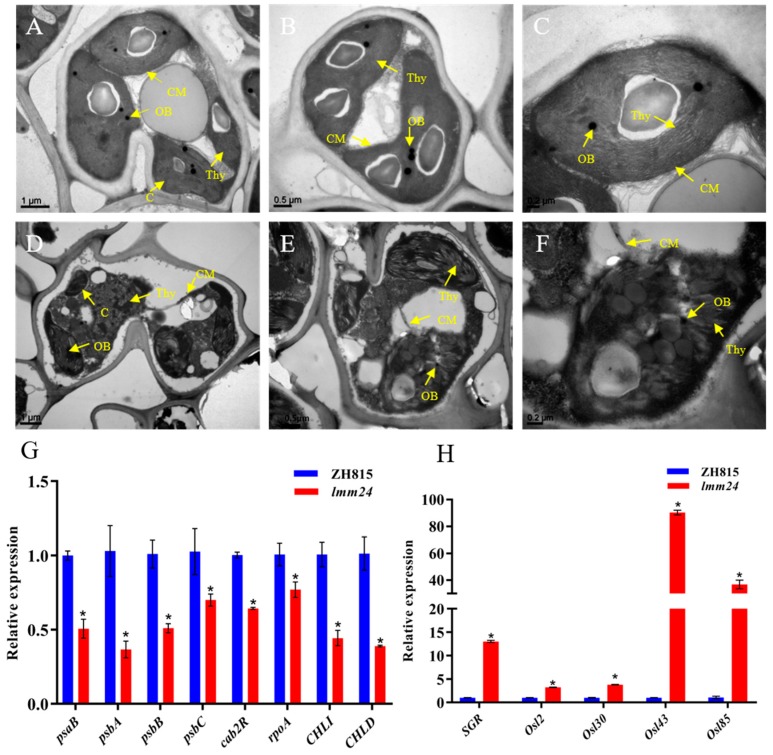
Qualitative TEM observation and genes expression analysis. (**A**–**C**) Observation of chloroplasts in ZH8015 mesophyll cells by TEM. C: chloroplast; Thy: thylakoid lamellae; OB: osmiophilic body; CM: chloroplast membrane. (**D**–**F**) Observation of chloroplasts in *lmm24.* (**G**) Expression levels of photosynthesis-related genes. *β-actin1* as a reference gene. (**H**) Expression levels of *SGR* gene and senescence-associated genes, *β-actin1* as a reference gene. The expression level of each gene in ZH8015 was normalized to 1. Data are means ± SE of three biological replicates. The *p*-value is calculated by the Mann–Whitney U test method. **p* < 0.01.

**Figure 4 ijms-20-03243-f004:**
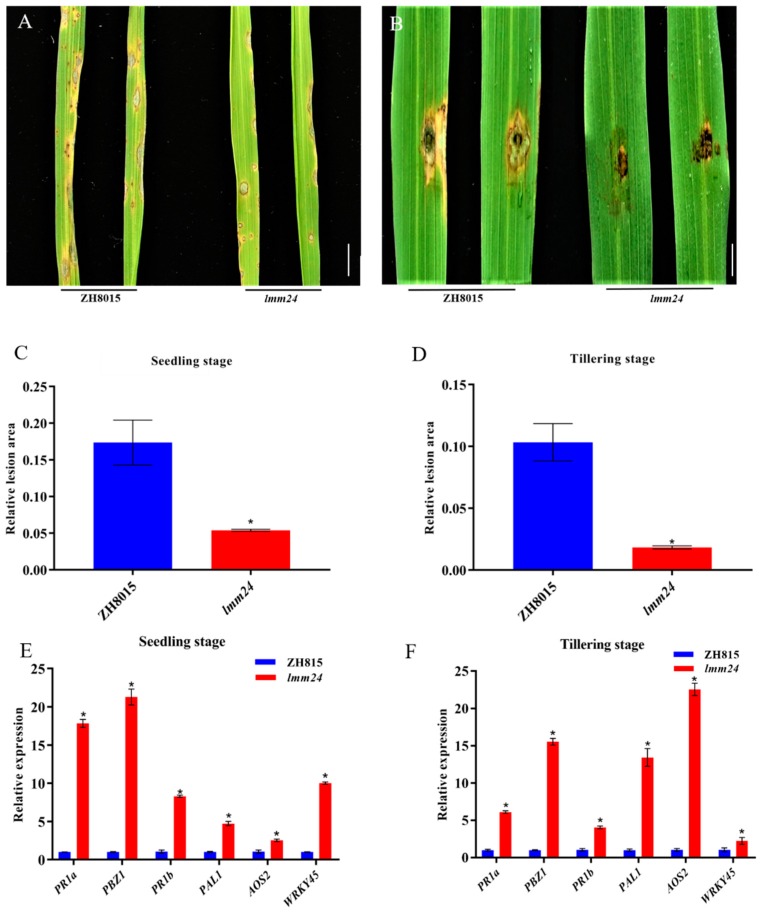
Enhanced resistance in the *lmm24* to *M. oryzae*. (**A**) Spraying method to infect ZH8015 and *lmm24* at seedling stage (20 dps), leaves were photographed at 7 dpi. (**B**) Punch inoculation method to infect ZH8015 and *lmm24* at tillering stage (50 dps), leaves were photographed at 12 dpi. (**C**) Relative lesion area of the inoculated leaves of ZH8015 and *lmm24* plants at seedling stage (7 dpi). Data are means ± SE of 10 plants (Student’s *t*-test: ***p* < 0.01) (**D**) Relative Lesion area of the inoculated leaves of ZH8015 and *lmm24* plants at tillering stage (12 dpi). Data are means ± SE of 10 plants (Student’s *t*-test: **p* < 0.01). (**E**) Expression levels of PR genes in leaves of ZH8015 and *lmm24* plants at seedling stage (20 dps). *β-actin1* as a reference gene. (**F**) Expression levels of PR genes in leaves of ZH8015 and *lmm24* plants at tillering stage (50 dps), *β-actin1* as a reference gene. the expression level of each gene in ZH8015 was normalized to 1. Data are means ± SE of three biological replicates. The *p*-value is calculated by the Mann–Whitney U test method. **p* < 0.01. Scale bar: 3 cm in (**A**), 2 cm in (**B**).

**Figure 5 ijms-20-03243-f005:**
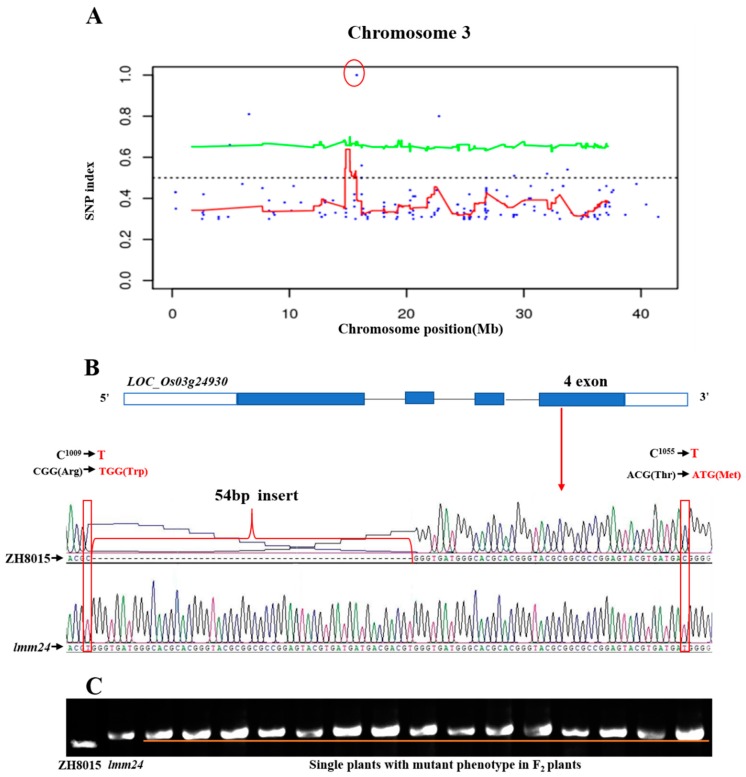
Candidate region and structure of *LMM24*. (**A**) SNP index Manhattan plot of chromosome 3. Candidate gene sites are shown in the red circle. (**B**) Structure and mutation site of *lmm24*. Four exons and three introns are indicated by blue rectangles and black lines. Two C to T point and 54 bp insertion mutation were identified in the fourth exon. Sequence analysis of the mutation site in plants of ZH8015 and *lmm24*. (**C**) Agarose gel electrophoresis using a molecular marker at the insertion site. Agarose is 2%. The fragment corresponding to ZH8015 is the size of the PCR product without insertion mutation. The fragment corresponding to *lmm24* is the size of the PCR product with the insertion mutation.

**Figure 6 ijms-20-03243-f006:**
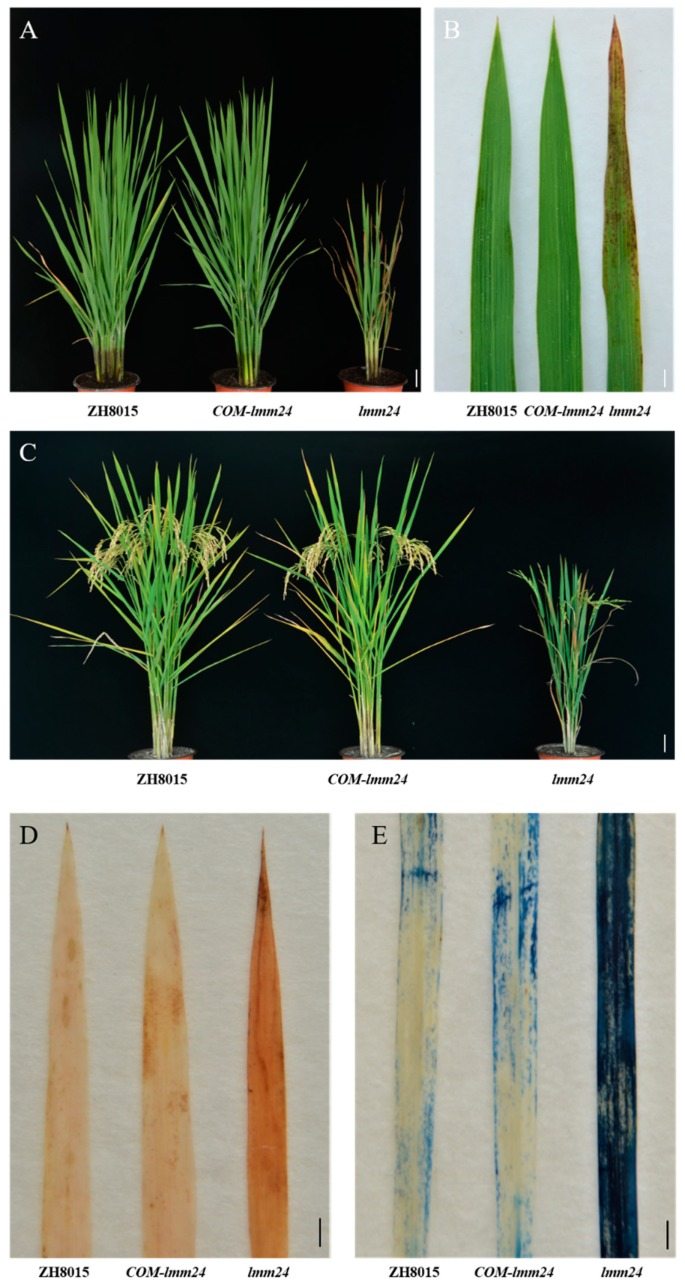
Genetic complementation of *LMM24*. (**A**) *lmm24* plant transformed with the genomic sequence of ZH8015 (*COM-lmm24*) was completely recovered to the ZH8015 phenotype at tillering stage. (**B**) Leaves phenotype of ZH8015, transgenic plant and *lmm24*. (**C**) The phenotype of ZH8015, transgenic plant and *lmm24* at harvest period. (**D**) DAB staining experiment. (**E**) Evans blue staining analysis. Scale bar: 10 cm in (**A**,**C**), 2 cm in (**B**),1 cm in (**D**,**E**).

**Figure 7 ijms-20-03243-f007:**
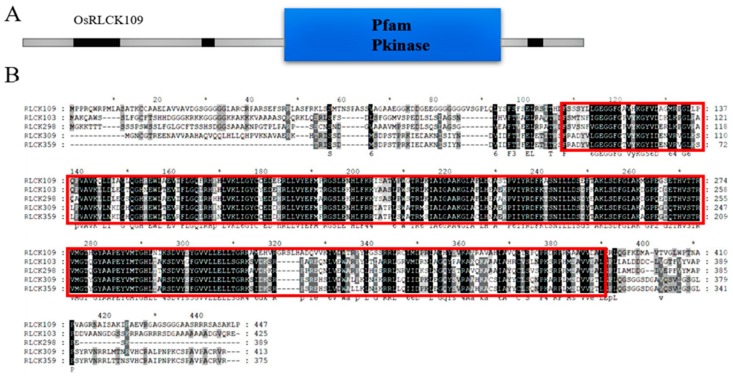
Predicted domains of OsRLCK109 and multiple sequence alignment. (**A**) Prediction of OsRLCK109 protein domains on smart website, blue rectangular region indicates the kinase domain. (**B**) Protein sequence alignment of OsRLCK109 and four other RLCKs, the highly conserved kinase domain is shown in the red box.

**Figure 8 ijms-20-03243-f008:**
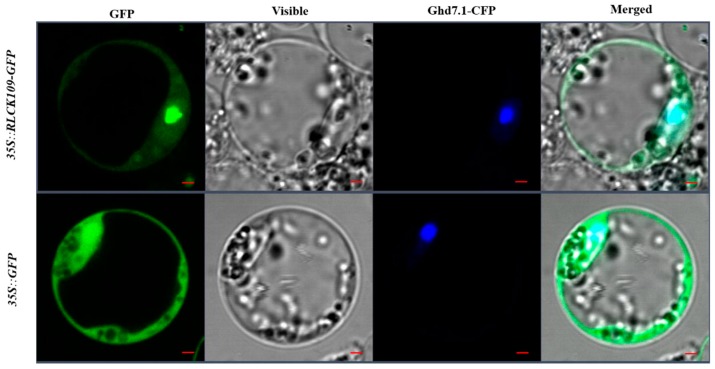
Subcellular localization of OsRLCK109. *35S::RLCK109-GFP*, RLCK109 fused to GFP; *35S::GFP*, empty GFP; Ghd7.1-CFP, nucleus marker. Scale bar: 2 μm.

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
