# Peer review of "LMM24 Encodes Receptor-Like Cytoplasmic Kinase 109, Which Regulates Cell Death and Defense Responses in Rice"

_ijms, 2019, doi:10.3390/ijms20133243_

Reviewer 1 Report

Mutations in LMM genes generate spontaneous lesions similar to those of the biotic or environmental stresses. This response is linked to a heterochronic activation of cell death so those genes are instrumental to better understand how this important process is regulated.

Zhang and co-workers have identified through a mapping-by-sequencing approach one of those genes in rice, LMM24. Mutations in this gene cause defects in the general growth of the plant and also alter other important agronomic traits. The authors showed how this phenotype is correlated with higher amount of ROS, cell death, and leaf senescence. At the cellular level, authors show some chloroplast defects in the mutant line and how this is correlated with a downregulation of nuclear photosynthetic genes. Moreover, they show how this mutation confers resistance to a fungal infection. Finally, they perform a subcellular localization experiment using protoplasts that indicates that LMM24 works in the nucleus and in the cytoplasm.

Although it is largely descriptive, it is an interesting piece of work for those working in the field and, in general, the conclusions are consistent with the observations. I think that the manuscript could be accepted but there are a relatively high number of issues that must be addressed:

Major points:

Line 82: In a classical genetic screen, at the beginning of the results part, it must be explained the origin of the mutant and what was the initial interest of the mutagenesis. That way it would easy to understand that ZH8015 is the wild-type parental strain.

Considering that only one allele of the gene is reported, additional analysis must be performed using the complementation line. The complementation line is only showed how restores the leaf and size phenotypes but nothing is reported regarding the agronomic traits showed in Figure S1, the accumulation of ROS and cell death (Figure 2), the morphology and transcriptional state of the chloroplast (Figure 3) or the resistance to fungal infection (Figure 4).

From my view, although there are reasons to perform the experiments introduced in lines 92 (effects of light), 109 (ROS and cell death) and 168 (mapping) there is a clear lack of justification in the manuscript. At least one or two sentences must be added (similarly to what authors do in lines 124 and 143).

Line 167: the description, justification and explanation of this part are far of being optimal. First, I would consider moving this part to just below the mutant and phenotype description. Second, authors are using a Mapping-by-sequencing approach but they do not even mention it. I don’t see why they have obtained such a big different read depth between wild-type and mutant populations. The data showed in Figure S3 must be rearranged: first two tables must be transformed into Table S4 and Table S5. The information given by the last two columns in the first table to me is meaningless. Any nucleotide covered only by 1 or 4 reads is not reliable. The remaining information would be more understandable if it would be presented as a chart instead of tables. Regarding this experiment, Figure S2 is not useful at all. This is a classic mapping-by-sequencing approach that has been used in many (plant) species. As there is not a particular design for this paper, I would suppress this figure. Instead, as I suggested above, I would expand the explanation. Finally, I see the Manhattan plot in Figure 5 quite blurred. Maybe, an indication of what region from A is analyzed in B would be interesting. The raw data must be deposited in a public repository (like SRA) and the reference number of the data must be included in the final version of the publication.

Line 202: How many plants were obtained in the complementation experiments must be included in the manuscript.

Line 218: authors do not show any protein structure. The sentence should be changed to: “The predicted protein scheme for OsRLCK109 shows a kinase domain…”

The subcellular localization experiment: I am not convinced about the use of “free” mCherry as a marker of anything. I suggest the use of DAPI to clearly demonstrate the partially nuclear localization of the protein. Moreover, I think that the use of the 35S promoter must be indicated in the figure. Finally, in the methods section, it must be indicated what lamp or laser was used to excite the fluorescent proteins and how was detected the signal (filters).

Line 247, I think in the discussion should be highlighted the fact that a mutant generated with a chemical mutagen (EMS) harbours 2 different transitions and one relatively big deletion in the same gene. Similar cases should be cited here.

Minor points:

Line 40: In this sentence, there is an enumeration of plant species by their common names. It should be changed. At least the first time authors name a plant species must be done using the scientific name.

Line 41: Here authors say that there are more than 20 LMM genes but they only provide one reference, the number 12. This journal does not have any limitation on the number of references. Although they later explain some examples, authors should provide all the relevant references here.

Line 46: “jasmonate (JA)” should be substituted by “jasmonic acid (JA)”

Line 49: SDS must be written in italics as authors refer here to the gene.

Line 61: I would add “Lines expressing RNAi constructs against OsRLCK57…” and use italics when referring to genes.

Line 69: similar to my previous comment in line 41, given the lack of limits in the reference number, please cite the original works here instead of a review (25).

Line 70: authors here introduce how MutMap works and how it has been used for other genes in rice. However, the genetic basis of this analysis, the mapping-by-sequencing, has been discussed in many reviews and it is not even mentioned. Candela et al. (2015). Journal of Integrative Plant Biology 57, 606-612 is an example but there are many others.

Line 98: here authors list several agronomic traits. All of them should be mention, avoiding the use of “and other agronomic traits”. To avoid lack of consistency, in the title of Figure S1, authors should say ZH8015 instead of WT.

Line 114: A more detailed explanation about what a TUNEL assay is and how it is performed must be included here. Something similar to what is explained in line 320 would be enough.

Lines 124-134 and Figure 3: If authors have taken enough pictures and have analyzed enough leaves, it should be interesting to quantify some of the phenotypes they observe. If possible, it should be added to Figure 3. If they do not do that, the figure title should be changed to “Qualitative TEM observations and…”. Regarding the qRT-PCR analysis, it is well described in methods except that they do not explain how they perform the normalization. Do they use the 2-ddCt method or another? Why?. Most importantly, they explain that analyze 3 biological replicates in triplicate, what I consider is good enough, however, they use the Student t-test, which is a parametrical test that requires at least 10 values to be used correctly. I would suggest using a non-parametrical test, like the Mann-Whitney U test, which can be used with 9 values. This is also valid for Figure 4E and 4F.

Line 197: please include the percentage of Agarose used in this gel.

Line 199, 228 and Table S2: It must be indicated the name of the primers used in the construction of each transgene. On the other hand, Table S2 should be re-titled as these primers are used for PCR and for cloning.

Text comments:

Line 31. I would change “phenotype similar” for “similar phenotype”.

Line 50: the sentence “a lesion mimic phenotype develops…” is not very clear to me. I would rewrite it.

Lines 85, 102, 158, 159, 163, 164: Recurrent mistake: “15dps” must be replaced by “15 dps”. There must be a space between the number and the acronym dps.

Lines 106, 107, 122, 166, 211. Similar to the previous one but with the scale bars.

From my point of view, if in a particular Figure you are showing a statistical analysis and there is only one degree of significance, there should be only one asterisk, and the meaning of this asterisk should be explained in the legend. This is the case for Figure 3 and 4, but the authors use two asterisks instead. The only exception is Figure S1, where I agree, two asterisks should be applied.

Author Response

Response to Reviewer 1 Comments

Major points:

 Point 1: Line 82: In a classical genetic screen, at the beginning of the results part, it must be explained the origin of the mutant and what was the initial interest of the mutagenesis. That way it would easy to understand that ZH8015 is the wild-type parental strain.

 Response 1: Based on your comments, we added an introduction to the mutant material in the final part of the “Introduction”.

Point 2: Considering that only one allele of the gene is reported, additional analysis must be performed using the complementation line. The complementation line is only showed how restores the leaf and size phenotypes but nothing is reported regarding the agronomic traits showed in Figure S1, the accumulation of ROS and cell death (Figure 2), the morphology and transcriptional state of the chloroplast (Figure 3) or the resistance to fungal infection (Figure 4).

 Response 2: Your suggestion is reasonable. Because we are concerned about the recovery of lesions on the leaves, so that the identification of other phenotypes is neglected. At present, we only have plant materials in the seedling stage. In the additional materials, we supplement the identification of the accumulation of ROS and cell death. Identification of other traits may take at least three months, but our data can support our conclusions. We hope to get your understanding.

Point 3: From my view, although there are reasons to perform the experiments introduced in lines 92 (effects of light), 109 (ROS and cell death) and 168 (mapping) there is a clear lack of justification in the manuscript. At least one or two sentences must be added (similarly to what authors do in lines 124 and 143).

 Response 3: We accepted your suggestion and made changes in the manuscript.

Point 4: Line 167: the description, justification and explanation of this part are far of being optimal. First, I would consider moving this part to just below the mutant and phenotype description. Second, authors are using a Mapping-by-sequencing approach but they do not even mention it. I don’t see why they have obtained such a big different read depth between wild-type and mutant populations. The data showed in Figure S3 must be rearranged: first two tables must be transformed into Table S4 and Table S5. The information given by the last two columns in the first table to me is meaningless. Any nucleotide covered only by 1 or 4 reads is not reliable. The remaining information would be more understandable if it would be presented as a chart instead of tables. Regarding this experiment, Figure S2 is not useful at all. This is a classic mapping-by-sequencing approach that has been used in many (plant) species. As there is not a particular design for this paper, I would suppress this figure. Instead, as I suggested above, I would expand the explanation. Finally, I see the Manhattan plot in Figure 5 quite blurred. Maybe, an indication of what region from A is analyzed in B would be interesting. The raw data must be deposited in a public repository (like SRA) and the reference number of the data must be included in the final version of the publication.

 Response 4: 1. We think that 2.1-2.4 is part of the phenotypic analysis, so we put 2.5 behind.

2. The wild-type background is cleaner than the F2 plants, because of the mutations, there are many mutation sites in the F2 generation that are not related to the phenotype of the lesion. Increasing the depth of F2 generation sequencing is helpful to eliminate the interference of irrelevant SNPS on the results. pool-WT was 97.69% coverage of the rice genome, pool-M was 97.78% coverage, we are very confident about our data.

3. We have modified the problem with Figure S3.

4. Figure S2 has been deleted.

5. Manhattan plot is very blurry because it is too big, we put it in Supplementary information so that it can be clearly displayed.

6. ZH8015 is a recovery system selected by our laboratory. At present, we are re-sequencing the genome of ZH8015 and adding annotations. Before the results of the annotations are published, our sequencing data is confidential and we hope to get your understanding.

Point 5: Line 202: How many plants were obtained in the complementation experiments must be included in the manuscript.

 Response 5: Information has been added to the manuscript.

Point 6: Line 218: authors do not show any protein structure. The sentence should be changed to: “The predicted protein scheme for OsRLCK109 shows a kinase domain…”

 Response 6: We accepted your suggestion and made changes in the manuscript.

Point 7: The subcellular localization experiment: I am not convinced about the use of “free” mCherry as a marker of anything. I suggest the use of DAPI to clearly demonstrate the partially nuclear localization of the protein. Moreover, I think that the use of the 35S promoter must be indicated in the figure. Finally, in the methods section, it must be indicated what lamp or laser was used to excite the fluorescent proteins and how was detected the signal (filters).

 Response 7: Based on your suggestion we conducted a DAPI experiment, unfortunately we failed, in order to verify the nuclear localization of rlck109, we added nuclear marker proteins. Other information we have supplemented.

Point 8: Line 247, I think in the discussion should be highlighted the fact that a mutant generated with a chemical mutagen (EMS) harbours 2 different transitions and one relatively big deletion in the same gene. Similar cases should be cited here.

 Response 8: To be honest, EMS mutagenesis often results in single-base mutations, with fewer examples of large fragment insertions and single-base mutations. Due to the limited reading capacity of the author, there are no reports of the same type of mutation as this article.

Minor points

 Point 1: Line 40: In this sentence, there is an enumeration of plant species by their common names. It should be changed. At least the first time authors name a plant species must be done using the scientific name.

 Response 1: We accepted your suggestion and made changes in the manuscript.

Point 2: Line 41: Here authors say that there are more than 20 LMM genes but they only provide one reference, the number 12. This journal does not have any limitation on the number of references. Although they later explain some examples, authors should provide all the relevant references here.

 Response 2: Because the article is not a review article, the author does not list all the cloned LMM genes, the author modified the expression in the manuscript, and deleted the reference.

Point 3: Line 46: “jasmonate (JA)” should be substituted by “jasmonic acid (JA)”

 Response 3: We accepted your suggestion and made changes in the manuscript.

Point 4: Line 49: SDS must be written in italics as authors refer here to the gene.

 Response 4: We accepted your suggestion and made changes in the manuscript.

Point 5: Line 61: I would add “Lines expressing RNAi constructs against OsRLCK57…” and use italics when referring to genes.

 Response 5: We accepted your suggestion and made changes in the manuscript.

Point 6: Line 69: similar to my previous comment in line 41, given the lack of limits in the reference number, please cite the original works here instead of a review (25).

 Response 6: We accepted your suggestion and made changes in the manuscript.

Point 7: Line 70: authors here introduce how MutMap works and how it has been used for other genes in rice. However, the genetic basis of this analysis, the mapping-by-sequencing, has been discussed in many reviews and it is not even mentioned. Candela et al. (2015). Journal of Integrative Plant Biology 57, 606-612 is an example but there are many others.

 Response 7: We accepted your suggestion and made changes in the manuscript.

Point 8: Line 98: here authors list several agronomic traits. All of them should be mention, avoiding the use of “and other agronomic traits”. To avoid lack of consistency, in the title of Figure S1, authors should say ZH8015 instead of WT.

 Response 8: We accepted your suggestion and made changes in the manuscript.

Point 9: Line 114: A more detailed explanation about what a TUNEL assay is and how it is performed must be included here. Something similar to what is explained in line 320 would be enough.

 Response 9: We added a description of the TUNEL assay in the manuscript.

Point 10: Lines 124-134 and Figure 3: If authors have taken enough pictures and have analyzed enough leaves, it should be interesting to quantify some of the phenotypes they observe. If possible, it should be added to Figure 3. If they do not do that, the figure title should be changed to “Qualitative TEM observations and…”. Regarding the qRT-PCR analysis, it is well described in methods except that they do not explain how they perform the normalization. Do they use the 2-ddCt method or another? Why?. Most importantly, they explain that analyze 3 biological replicates in triplicate, what I consider is good enough, however, they use the Student t-test, which is a parametrical test that requires at least 10 values to be used correctly. I would suggest using a non-parametrical test, like the Mann-Whitney U test, which can be used with 9 values. This is also valid for Figure 4E and 4F.

 Response 10: The figure title has been changed to “Qualitative TEM observations and…”; For the results of qPCR, we used the 2–ΔΔCT method. In addition, in order to clearly show the change of lmm24 relative to ZH8015, the value of ZH8015was normalized to 1. According to your suggestion, we changed the t-test to Mann-Whitney U test. There is no difference in the significance between the two methods. We modified it in the figure.

Point 11: Line 197: please include the percentage of Agarose used in this gel.

 Response 11: We added a description of the figure5D in the manuscript.

Point 12: Line 199, 228 and Table S2: It must be indicated the name of the primers used in the construction of each transgene. On the other hand, Table S2 should be re-titled as these primers are used for PCR and for cloning.

 Response 12: We have indicated the use of primers in the materials and methods. (4.3/4.8).

We re-titled the TableS2.

Text comments:

 Point 1: Line 31. I would change “phenotype similar” for “similar phenotype”.

 Response 1: We accepted your suggestion and made changes in the manuscript.

Point 2: Line 50: the sentence “a lesion mimic phenotype develops…” is not very clear to me. I would rewrite it.

 Response 2: We made changes in the manuscript.

Point 3: Lines 85, 102, 158, 159, 163, 164: Recurrent mistake: “15dps” must be replaced by “15 dps”. There must be a space between the number and the acronym dps.

 Response 3: We made changes in the manuscript.

Point 4: Lines 106, 107, 122, 166, 211. Similar to the previous one but with the scale bars.

 Response 4: We made changes in the manuscript.

Point 5: From my point of view, if in a particular Figure you are showing a statistical analysis and there is only one degree of significance, there should be only one asterisk, and the meaning of this asterisk should be explained in the legend. This is the case for Figure 3 and 4, but the authors use two asterisks instead. The only exception is Figure S1, where I agree, two asterisks should be applied.

 Response 5: We made changes in the manuscript.

 Reviewer 2 Report

In the manuscript “LMM24encodes receptor-like cytoplasmic kinase 109, which regulates cell death and defense responses in rice” the authors indicate the lmm24is a mutant of OsRLCK109 gene and use Lesion mimic mutants lmm24to study the molecular mechanism of cell death and immune response in rice. The authors use molecular, genetic and histological methods to characterize the lmm24mutant and identify their function on cell death and defense response. The data generated are reasonable given the conclusion of the paper. The study provides some new insights into complex defense pathways in plants.I think the paper is eligible for publication with some corrections. Specific comments on the manuscript are listed below:

 1.   I encourage the authors to have a native English speaker read and edit their manuscript to correct grammatical and tense errors and improve overall clarity of the manuscript.

2.   Page 2 line 80 Maybe add a little bit introduction about lmm24

3.   Page 2 line 87 “Some of the leaves withered and died because of accumulation of many lesions” there is no clear data indicated that, please explain it or revise it.

4.   Page 3 figure 1F and 1G, the authors say grain width and grain length of the lmm24 mutant is significantly decreased compared to ZH8015. However, it looks like there are not in the figure with a bar graph. The author may replace the bar graph with a box plot to show more detail about data or change with other traits. 

5.   Page 3 figure 1C, the authors may retake the image of ZH8015 and mutant lmm24 with a darker background to make sure the figure background is consistent.

6.   Page 4 figure 2C TUNEL assay to detect of DNA fragment in lmm24 and ZH8015. Apparently, the light intensity of ZH8015 and lmm24 images are different. Something related to laser intensity, etc. Please clarify it. Figure legend needs to rewrite because of grammar errors.

7.   Page 9 figure 5D, agarose gel electrophoresis image need some explanation. What are those two different sizes of DNA bands? Molecular markers (FP+RP, LMM24-1F/R and LMM24-2F/R. etc.) Please mark in the image or explain in the figure legend.

8.    Method sections need to provide more detailed information for your audience to repeat your experiment. Section 4.1 humidity in the light chamber is not listed; Section 4.2, are there any available link that reader can get access to original Illumina HiSeqTM dataset. Section 4.3, please include all the reagent that you used and more detail about Agrobacterium transformation and following steps.

9.   Page 14, section 4.6 RNA isolation and Q-RT-PCR analysis need to rewrite for publication. Some steps are missed and Q-RT-PCR needs at least 2 reference gene, etc. Details for Q-RT-PCR data submission guideline refer to MIQE guideline in PUBMED.

10.Page 14 line 337, method section 4.8 subcellular localization: detail information is missed for the confocal microscope. (laser emission and excitation wavelength, etc.)

11.Discussion sections page 12-13: This discussion section is very brief. A more thoughtful and deeper synthesis of the data and discussion of its relevance to other researches would be helpful. For example how mutant lmm24 or OsRLCK109 related to ROS-dependent cell depth or immune response in other research and their links to complex immune response pathways?

  Author Response

Response to Reviewer 2 Comments

 Point 1: I encourage the authors to have a native English speaker read and edit their manuscript to correct grammatical and tense errors and improve overall clarity of the manuscript.

 Response 1: We have already polished the manuscript before submitting the manuscript. We accepted your comments and we retouched the manuscript. The editor only made minor changes to the manuscript.

Point 2: Page 2 line 80 Maybe add a little bit introduction about lmm24

 Response 2: We accepted your suggestion and made changes in the manuscript.

Point 3: Page 2 line 87 “Some of the leaves withered and died because of accumulation of many lesions” there is no clear data indicated that, please explain it or revise it.

 Response 3: In the manuscript we removed this description.

Point 4: Page 3 figure 1F and 1G, the authors say grain width and grain length of the lmm24 mutant is significantly decreased compared to ZH8015. However, it looks like there are not in the figure with a bar graph. The author may replace the bar graph with a box plot to show more detail about data or change with other traits.

 Response 4: Although there is no significant difference in the figure, there are significant differences in the statistics of Figure S1. We changed Figure 1 to a box plot.

Point 5: Page 3 figure 1C, the authors may retake the image of ZH8015 and mutant lmm24 with a darker background to make sure the figure background is consistent.

 Response 5: In the experiment, we tried to shoot figure 1C with a black background, but the white background is more helpful for expressing the lesions of the leaves.

Point 6: Page 4 figure 2C TUNEL assay to detect of DNA fragment in lmm24 and ZH8015. Apparently, the light intensity of ZH8015 and lmm24 images are different. Something related to laser intensity, etc. Please clarify it. Figure legend needs to rewrite because of grammar errors.

 Response 6: The light intensity of our photos is the same. The difference in the light intensity reflected in the pictures may be due to the different treatment effects of reagents on different samples. In order to avoid disputes, we have changed the photos.

Point 7: Page 9 figure 5D, agarose gel electrophoresis image need some explanation. What are those two different sizes of DNA bands? Molecular markers (FP+RP, LMM24-1F/R and LMM24-2F/R. etc.) Please mark in the image or explain in the figure legend.

 Response 7: We accepted your suggestion and made changes in the manuscript.

Point 8: Method sections need to provide more detailed information for your audience to repeat your experiment. Section 4.1 humidity in the light chamber is not listed; Section 4.2, are there any available link that reader can get access to original Illumina HiSeqTM dataset. Section 4.3, please include all the reagent that you used and more detail about Agrobacterium transformation and following steps.

 Response 8: We added some content to the manuscript. ZH8015 is a recovery system selected by our laboratory. At present, we are re-sequencing the genome of ZH8015 and adding annotations. Before the results of the annotations are published, our sequencing data is confidential and we hope to get your understanding.

Point 9: Page 14, section 4.6 RNA isolation and Q-RT-PCR analysis need to rewrite for publication. Some steps are missed and Q-RT-PCR needs at least 2 reference gene, etc. Details for Q-RT-PCR data submission guideline refer to MIQE guideline in PUBMED.

 Response 9: We added some content to the manuscript

Point 10: Page 14 line 337, method section 4.8 subcellular localization: detail information is missed for the confocal microscope. (laser emission and excitation wavelength, etc.)

 Response 10: We added some content to the manuscript.

Point 11: Discussion sections page 12-13: This discussion section is very brief. A more thoughtful and deeper synthesis of the data and discussion of its relevance to other researches would be helpful. For example how mutant lmm24 or OsRLCK109 related to ROS-dependent cell depth or immune response in other research and their links to complex immune response pathways?

 Response 11: We added some content to the manuscript.

Round  2

Reviewer 1 Report

In this revised version of the manuscript, the authors have followed many of my suggestions. However, there are still some important issues that have to be addressed:

·           In general, I consider the English level quite low. A review by an English native-speaker would improve a lot this article.

·           Lines 17, 76 and 201: MutMap is a software, not a method. The gene identified in this paper has been found using a mapping-by-sequencing strategy. This method that, according to Response 4.2, authors do not understand, is based on linkage disequilibrium. There are plenty of reviews about this and I think it should be better explained, at least, in the introduction.

·           Line 41, authors did not follow my suggestion of using scientific names of plant species at least once. They did it only for Arabidopsis, but not for maize, rice, wheat and barley. In the same line, LMM should be in italics.

·           Line 93: LMM24 should be in italics.

·           Figure 1C: I would indicate the panel “C” in black instead of white.

·           Line 132: Trying to explain what a TUNEL assay is, authors, explain the scientific basis behind the assay but they did not mention that this is a TUNEL assay.

·           Line 162: Authors should indicate what ACTIN gene was used. There are several.

·           Lines 164, 187 and 192: authors corrected most of these mistakes but there still some. “7dpi” must be changed to “7 dpi” and the same for “mean±SE”

·           Figure 5: It must be indicated that the Manhattan plot corresponds to chromosome 3.

·           Line 236: “6A, B,C” must be changed to “6A, B, C”

·           Line 272: I would express this as “protein is localized mainly in the nucleus but there is also a weaker signal from cytoplasm” .

·           Line 260: Figure 7 title. The figure does not show any structure.

·           Lines 371 and 377: spelling mistakes.

·           Line 397: Omitting my suggestion, the authors are not explaining what lasers or lamps were used to excite the fluorescent proteins.

·           Table S1: I would say “wild-type” instead of “normal” plants.

·           Table S5: I think there is a mistake with the spelling of ACTIN.

·           Figure S4: I suggest to turn right all the Manhattan plots. I also suggest expanding the figure legend that in the current status does not explain anything.

Author Response

Response to Reviewer 1 Comments

 Point 1: In general, I consider the English level quite low. A review by an English native-speaker would improve a lot this article.

 Response 1: We have extensive English editing of our manuscripts.

Point 2: Lines 17, 76 and 201: MutMap is a software, not a method. The gene identified in this paper has been found using a mapping-by-sequencing strategy. This method that, according to Response 4.2, authors do not understand, is based on linkage disequilibrium. There are plenty of reviews about this and I think it should be better explained, at least, in the introduction.

 Response 2: We can't understand your comment: MutMap is a software, not a method. MutMap is a new forward genetic approach based on high-throughput next-generation sequencing technologies, has been discussed in many reviews. Abe et al. (2012). Nature Biotechnology 2012, 30, (2), 174-178 is an example. According to Candela's description, mapping-by-sequencing strategy is a summary of case studies where the use of NGS technologies has led to the identification of point mutations. There is no doubt that MutMap is a kind of mapping-by-sequencing strategy. It is more accurate to describe our method as MutMap, because mapping-by-sequencing strategy contains multiple methods.

Point 3: Line 41, authors did not follow my suggestion of using scientific names of plant species at least once. They did it only for Arabidopsis, but not for maize, rice, wheat and barley. In the same line, LMM should be in italics.

 Response 3: We accepted your suggestion and made changes in the manuscript.

Point 4: Line 93: LMM24 should be in italics.

 Response 4: We accepted your suggestion and made changes in the manuscript.

Point 5: Figure 1C: I would indicate the panel “C” in black instead of white.

 Response 5: We accepted your suggestion and made changes in the manuscript.

Point 6: Line 132: Trying to explain what a TUNEL assay is, authors, explain the scientific basis behind the assay but they did not mention that this is a TUNEL assay.

 Response 6: We accepted your suggestion and made changes in the manuscript.

Point 7: Line 162: Authors should indicate what ACTIN gene was used. There are several.

 Response 7: Our description wasn't clear enough. We've revised it in the manuscript.

Point 8: Lines 164, 187 and 192: authors corrected most of these mistakes but there still some. “7dpi” must be changed to “7 dpi” and the same for “mean±SE”

 Response 8: We accepted your suggestion and made changes in the manuscript.

Point 9: Figure 5: It must be indicated that the Manhattan plot corresponds to chromosome 3.

 Response 9: We have made some modifications in the picture.

Point 10: Line 236: “6A, B,C” must be changed to “6A, B, C”

 Response 10: We accepted your suggestion and made changes in the manuscript.

Point 11: Line 272: I would express this as “protein is localized mainly in the nucleus but there is also a weaker signal from cytoplasm” .

 Response 11: We accepted your suggestion and made changes in the manuscript.

Point 12: Line 260: Figure 7 title. The figure does not show any structure.

 Response 12: We modified the title of the Figure 7.

Point 13: Lines 371 and 377: spelling mistakes.

 Response 13: We accepted your suggestion and made changes in the manuscript.

Point 14: Line 397: Omitting my suggestion, the authors are not explaining what lasers or lamps were used to excite the fluorescent proteins.

 Response 14: We accepted your suggestion and made changes in the manuscript.

Point 15: Table S1: I would say “wild-type” instead of “normal” plants.

 Response 15: We accepted your suggestion and made changes in the manuscript.

Point 16: Table S5: I think there is a mistake with the spelling of ACTIN.

 Response 16: We accepted your suggestion and made changes in the manuscript.

Point 17: Figure S4: I suggest to turn right all the Manhattan plots. I also suggest expanding the figure legend that in the current status does not explain anything.

 Response 17: We accepted your suggestion and made changes in the manuscript.